# Psychometric Properties of the Greek Version of the Medical Office on Patient Safety Culture in Primary Care Settings

**DOI:** 10.3390/medicines8080042

**Published:** 2021-07-26

**Authors:** Ioannis Antonakos, Kyriakos Souliotis, Theodora Psaltopoulou, Yannis Tountas, Athanasios Papaefstathiou, Maria Kantzanou

**Affiliations:** 1Faculty of Medicine, National and Kapodistrian University of Athens, Rimini 1 Chaidari, 124 62 Athens, Greece; 2Faculty of Social & Political Sciences, University of Peloponnese, 201 00 Corinth, Greece; ksouliotis@uop.gr; 3Laboratory of Hygiene, Epidemiology and Medical Statistics, University of Athens Medical School, 115 27 Athens, Greece; tpsaltop@med.uoa.gr (T.P.); ytountas@med.uoa.gr (Y.T.); mkatzan@med.uoa.gr (M.K.); 4Endocrine and Metabolic Bone Disorders Unit, 2nd Department of Internal Medicine and Research Institute and Diabetes Center, Attikon University Hospital, 124 62 Athens, Greece; athpap@med.uoa.gr

**Keywords:** patient safety culture, factor analysis, primary care

## Abstract

Background: Safety culture is considered one of the most crucial premises for further development of patient care in healthcare. During the eight-year economic crisis (2010–2018), Greece made significant reforms in the way the primary health care system operates, aiming at the more efficient operation of the system without degrading issues of safety and quality of the provided health services. In this context, this study aims to validate a specialized tool—the Medical Office Survey on Patient Safety Culture (MOSPSC)—developed by the Agency for Healthcare Research and Quality (AHRQ) to evaluate primary care settings in terms of safety culture and quality. Methods: Factor analysis determined the correlation of the factor structure in Greek data with the original questionnaire. The relation of the factor analysis with the Cronbach’s coefficient alpha was also determined, including the construct validity. Results: Eight composites with 34 items were extracted by exploratory factor analysis, with acceptable Cronbach’s alpha coefficients and good construct validity. Consequently, the composites jointly explained 62% of the variance in the responses. Five items were removed from the original version of the questionnaire. As a result, three out of the eight composites were a mixture of items from different compounds of the original tool. The composition of the five factors was similar to that in the original questionnaire. Conclusions: The MOSPSC tool in Greek primary healthcare settings can be used to assess patient safety culture in facilities across the country. From the study, the patient safety culture in Greece was positive, although few composites showed a negative correlation and needed improvement.

## 1. Introduction

A diagnostic error is defined as a “failure to create an accurate and timely explanation of the patient’s health problem or communicate that explanation to the patient”, according to a recent Institute of Medicine (IOM) report titled “Improving Diagnosis in Health Care” [1]. There have also been other formal definitions of diagnostic error presented in the past [2,3,4].

The main characteristics of primary care (i.e., first contact care, continuity, comprehensiveness, and coordination) [5] make it a high-risk area for errors. Physicians are regularly confronted with large patient loads and are forced to make decisions in the face of uncertainty [6]. Undifferentiated presenting signs are the norm for both common and unusual diseases in primary care, which tend to be benign and self-limiting. The diagnosis usually takes place over a period of time and across numerous sessions of care [7,8]. Physicians must carefully weigh the danger of missing a serious illness against the wise use of sometimes limited and expensive referral and testing resources. As a result, diagnostic errors that result in patient damage due to incorrect or delayed testing or treatment have become a global safety issue [9].

Common diagnostic errors reported in a survey of primary care physicians included cancer, pulmonary embolism and coronary artery disease [10]. Another survey of US internal medicine physicians reported both outpatient and inpatient errors related to pulmonary embolism (4.5%), drug reactions (4.5%), lung, colorectal and breast cancers (3.9%, 3.3% and 3.1%, respectively), acute coronary syndrome (3.1%), and stroke (2.6%) [6]. In a US study of 181 malpractice claims, cancer was the most common diagnosis involved [11]. An analysis of 1000 negligent claims against the UK general practitioners identified diagnostic errors most commonly involving infections, trauma, and cancer [12]. Malpractice claims, however, tend to involve diagnoses that are more serious or most harmful if not diagnosed correctly in a timely fashion and do not necessarily represent error frequency.

Due to the errors identified in various health care facilities in the past, most health care organizations encourage countries to enact policies that concern patient safety. In particular, the Institute of Medicine requires that healthcare facilities ensure patient safety in all their activities [13]. Similarly, the European Council Recommendation on Patient Safety, Prevention, and Control set out the benefits of general safety measures as the sick are attended to [14]. In its report, the council stated that health facilities that do not observe patient safety incur various costs and put pressure on available resources [15]. On the other hand, health facilities can implement patient safety procedures, especially those with chronic conditions, such as diabetes and cancer. Patient safety programs [16,17] involve a range of activities beyond the medication that are channeled towards the recovery of patients in health facilities. Such programs should include the enhancement of safety culture among health care professionals. The Advisory Committee on the Safety of Nuclear Installations [18] provides the following definition of safety culture that can easily be adapted to the context of patient safety in health care: ‘‘The safety culture of an organization is the product of individual and group values, attitudes, perceptions, competencies, and patterns of behavior that determine the commitment to, and the style and proficiency of, an organization’s health and safety management. Medical practitioners in all facilities should be aware and implement these values and beliefs that ensure the wellbeing of patients, which leads to positive outcomes [19]. In addition, the Patient Safety programs should involve leaders and management to establish effective ways of promoting the culture in health facilities [20]. In this context various tools and quantitative instruments have been designed to assess healthcare settings in terms of safety and quality especially in hospital environment [21]. In recent years, concerns have been raised about the medical errors that have occurred in primary health care and the need to create a safer environment for patients [22,23,24].

A few articles have been published utilizing tools that assess patient safety culture in primary care settings [25,26,27,28,29,30,31]. Greece was under the supervision of the International Monetary Fund from 2010 to 2018. In these eight years, primary health care has undergone many reforms. The most important of these reforms was the establishment of a single primary health care provider (PEDY), family doctor, and electronic prescribing [32]. Therefore, it is challenging to investigate the level of primary care services provided in terms of quality and safety using a specialized and validated questionnaire. This kind of study took place for the first time in Greek primary units and mainly aims to evaluate the psychometric properties of the translated Greek version of the Hospital Survey on Patient Safety Culture (G-MSOPSC) of the Agency for Healthcare Research and Quality (AHRQ) in the Greek primary healthcare settings [33].

Additional objects of the study are to investigate the degree of patient safety and the quality of health services in primary care settings in Greece, as well as the strong and weak areas of safety culture, regarding health professionals’ views.

## 2. Methods

### 2.1. Data Source

A cross-sectional, multicenter, multidisciplinary study was carried out to examine the psychometric properties of the MOSPSC survey. The survey took place at primary health care units (PHUs) of the first health region of Greece, located in the biggest prefecture of Greece, Attica, with 2,500,000 residents. Data collection took place from December 2019 to April 2020. Twelve PHUs were selected representative according to the type of medical services provided. The majority was medical services, 35 (52%), concerning the pathological sector, pediatrics, etc., followed by microbiology, 22 (33%), and the radiology department 10 (15%).

### 2.2. Instrument

The MOSPSC tool [34] includes 38 items that make up 10 composite measures of patient safety culture (Table 1). In addition, the survey included items regarding the “Patient Safety and Quality Issues” (9 items), “Information Exchange with Other Settings” (4 items), and “Average Overall Ratings on Quality and Patient Safety” (6 items). For positively worded items with 5-point response scales, percent positive response is the combined percentage of respondents within a PHU who answered, “Strongly agree” or “Agree,” or “Always” or “Most of the time,” depending on the response categories used for the item. The corresponding percentage for negatively worded items is the combined percentage of respondents within a PHU who answered “Strongly disagree” or “Disagree,” or “Never” or “Rarely”, because a negative answer on a negatively worded item indicates a positive response. Percent positive scores for the “Patient Safety and Quality Issues” items, as well as the “Information Exchange with Other Settings”, were calculated differently than the other survey items. The percent positive score for these 13 items is the sum of the three response options that represent the smallest frequency of occurrence. The reverse questions of the tool refer to items C3, C6, C8, C10, C12, C14, D4, D7, D10, E1, E2, E4, F3, F4, and F6. 

### 2.3. Translation Process

Concerning the translation process, after the permission obtained by the authors, MOSPSC was translated into Greek and then back into English by two independent researchers to ensure conformity of the translation. Subsequently, before the tool was used, it was handed to 35 health care professionals, first to ascertain that all components in the MOSPSC were understood. The questions in the sample were not included in the main MOSPSC survey to avoid repetition of responses. The Cronbach’s alpha formula was applied to test the reliability of the test, where the score was recorded as 0.8, which is a good indicator of the method’s scale.

### 2.4. Sample

To perform data collection, initially, the researcher contacted the PHU managers’ in order to inform them regarding issues of the research, such as objective, justification, risks, and benefits, as well as legal and ethical issues. After agreeing to participate, they received an envelope containing questionnaires that were shared and completed by the staff, accompanied by the Free and Informed Consent Term (FICT), in two copies. Respondents’ privacy was assured. A cross-sectional study was carried out in 12 primary health care facilities of the first health region of Attica that were selected in a representative way in terms of services provided and staff (Table 2). Data collection took place from December 2019 to May 2020. Although there was a lockdown period in Greece between 23 March 2020 and 6 May, the vast majority of the questionaries (444 or 97%) was already collected; so there was no bias due to the lockdown period.

Inclusion criteria for the responders were: being a professional of the multidisciplinary team that provided direct and indirect assistance to the patient, working in the unit for at least 30 days, and working at least 20 h per week. Responders who did not meet the above criteria were excluded. After applying these criteria, 770 professionals participated (response rate = 59.6%). Subsequently, 160 questionnaires were excluded for the following reasons: 70 respondents answered a portion of the questionnaire, 30 questionnaires had the same answer choice over and over again, and 60 responses were inconsistent. Finally, 151 respondents who had a part time job were excluded, resulting in a sample of 459 questionnaires being retained for further analysis (Figure 1).

### 2.5. Statistical Analysis

A factory analysis (FA) was carried out (principal axis extraction method, Varimax rotation [16] in order to prove that the current scales/dimensions may be fairly employed within the Greek context. The analyses were performed using SPSS (version 23.0; IBM SPSS, Armonk, NY, USA). A *p*-value < 0.05 was considered significant. When proving the number of elements, the eigenvalue (eigenvalue > 1: Kaiser’s criterion) was taken into screen plot and the future outcome of interpreting the elements. Kaiser’s criterion is trustworthy in a specimen of more than 250 respondents and when the average communality is greater than or equal 0.6. The figure of the screen plot supplies dependable knowledge when the sample is larger than 200 respondents [35].

### 2.6. Ethics

The research project was approved by the Research Ethics Committee of the Medical School of University of Athens and the Scientific Council of the first Health Region of Attica, respecting all ethical standards recommended by Greek law (IRB No. 076/25.02.2019).

## 3. Results

### 3.1. Sample

A total of 459 professionals participated in this study. Most respondents were nursing staff; 190 (41%), followed by physicians; 95 (21%); midwifes 80 (17%); clerical staff 55 (12%) and physical/occupational/therapists (9%). As to the professional and sociodemographic profile, 312 (68%) were women; 285 (62%) were aged between 30 and 49 years; 35 (72.3%) were married; 255 (55.6%) were working at the service for one to five years; and 204 (44.5%) were working for six years or more (Table 2).

### 3.2. Factor Analysis and Internal Consistency

The results of the factor analysis showed a significant adjustment of the G-MOSPSC scale represented by the Kais test (Kaiser–Meyer–Olkin) of 0.843 and the significant Bartlett sphericity test [c2(595) = 12,803; *p* < 0.001], which attested to the possibility of performing the factor analysis. The latent underlying criterion or eigenvalue was achieved, where only eigenvalues ≥1 were considered significant. The Guttman–Keiser criterion estimated eight latent variables should be extracted, where the first had an eigenvalue of 10.6, carrying about 30.2% variance, while in the last factor (factor 8) the eigenvalue was 1.14, which managed to explain 3.2% of variance. The factorial model reached a 77.6% explained variance ratio). Eight composites were obtained by factor analysis with 34 items (Table 3). Five out of 38 items of the original version of the tool (C1, D2, F2, D3, D9) did not have sufficient factor loading on any of the factors (all loadings < 0.50) or the cross loadings differ less than 0.2 and were eliminated. Table 3 also gives the mean scores with standard deviations and factor loadings per item. The eight composites in the G-MSOPSC had Cronbach’s coefficients between 0.70 and 0.88, which indicated good internal consistency of the Greek version of the questionnaire (Table 4). Comparing this structure with the one proposed by MOSPSC, “Communication About Error” gained several items from composites 1, 3, 6, and 8. Composites 5 and 9 did not suffer any changes and composites 2 and 4 lost two items (Table 4). The results were considered reliable because of the exploratory factor analysis model fit obtained through adequate free asymmetric distribution methods in order to estimate ordinal categorical items with nonparametric distribution.

### 3.3. Safety Culture Composite Measures and Overall Rating in Patient Safety and Quality

The most highly ranked composites by the respondents were “Teamwork” (82% positive rating), “Patient Care Tracking/Follow-up” (80%), “Organizational Learning” (80%), and “Overall Perception of Patient Safety and Quality” (78%). “Staff training” (70% of positive responses), “Communication About Errors” (70%), “Office Processes and Standardization” (67%), and “Communication Openness (64%) followed. The lowest scores were for “Owner/Managing Partner/Leadership Support for Patient Safety” (62%) and “Work Pressure and Pace” (46%) (Table 5).

Concerning perceptions of the staff about issues related to the quality of the services and patient safety, the assessed safety culture is good as the health care is equitable (85%), effective (75%), patient-centered (75%), efficient (63%), timely (65%), and safe (70%) (Table 6). These results seem to be very encouraging as they are slightly higher than 2020 AHRQ Database, although the sample size differs significantly [36].

## 4. Discussion

As far as we know, this is the first study conducted in Greece that reports on the structure, as well as the psychometric properties of G-MSOPSC in accordance with the guidelines of the AHRQ. Despite the fact that our results are aligned with the original version, concerning a-Cronbach’s values, some adaptations were demanded so that the Greek context is fitted correctly. An 8-factor model with 34 items performed better than the original one in the sample of the 12 Greek primary care units. Two out of eight composites remained the same as the original tool: “Staff Training” and “Office Processes and Standardization”. Items from different composites of the original tool created composites 1, 5, and 6 of the G-MOSPSC with high a-Cronbach’s values (0.97, 0.89 and 0.84, respectively). Similar results were obtained from the Portuguese [37] validation of the tool, where four out of ten composites were a mixture of different items of the original tool. The remaining three composites (2, 3, and 8) retained items from the following composites of the original tool: composite 2 from “Patient Care Tracking/Follow up” (D5, D6), composite 3 from “Overall Perceptions of Patient Safety and Quality (F4R, F3R, F6R)” and composite 8 from “Work Pressure and Pace (C11, C6R). The available evidence from studies conducted in European and non-European countries such as Spain [38], Portugal [37], Brazil [39], and Yemen [40] suggest adaptions to the US version of the tool. In Spain for instance, questions were added and, when assessing a-Cronbach for each dimension, unsatisfactory value was obtained for “Staff Training” and “Patient Care Tracking/Follow-up”. When validating for the Spanish version [38], the a-Cronbach ranged from 0.20 to 0.70, and “Information Exchange with Other Institutions” and “List of questions on patient safety and quality” due to high non-response and non-response rates were excluded. Similar results were found in the validation to Portuguese where a-Cronbach ranged from 0.52 to 0.88, and for the same reasons cited in the previous study both dimensions were excluded. An exception is the Brazilian version of the tool in which all the items and composites remained the same as the original version. Regarding the five items (C1, D2, F2, D3, D9) that were removed from the original tool due to the restrictions imposed by factor analysis, our belief is that these items should be kept since they signify important aspects of patient safety, as well as for comparative evaluation purposes of the tool. In addition, we kept the same structure of the questionnaire as the original one for purposes of benchmarking and foundation for improvement work since a-Cronbach’s values of the 10-composites G-MOSPSC questionnaire are appropriate and fully comparable to the original (Table 4). This study shows that the professionals interviewed had a positive safety culture (Table 5). The composite “Patient Care Tracking/Follow up” evaluated with 80% positive responses, while “Work Pressure and Pace” received the lowest percentage of positive responses (46%).

In Greece, the operation of primary healthcare units in small multidisciplinary teams appears to enhance employee cooperation. The high proportion linked with the “Teamwork” dimension, as well as the “Organizational Learning” dimension, reflect this. Inter-disciplinary cooperation appears to aid health professionals in comprehending shared contextual duties and obligations, allowing them to achieve organizational objectives, engage with and disseminate important information, and deliver safe and effective treatment. “Teamwork” emerged as the highest safety culture domain in Yemen (96%) [40] and in Holland (79.2%) [28].

Another significant aspect in the present study’s safety culture was “Patient Care Tracking/Follow-up.” This shows that PHU patients in Greece are reminded of their appointments, their commitment to the therapy process is confirmed, and the follow-up with patients who need monitoring is appropriate. This is primarily owing to the fact that primary care electronic systems have been updated in recent years.

Another strong area in the safety culture in the current study was the “Patient Care Tracking/Follow-up.” This indicates that PHU patients in Greece are reminded of their dates, their adherence to the therapeutic process is verified, and the follow-up with patients who require monitoring is adequate. This is largely due to the fact that electronic systems in primary care have been modernized in recent years (to incorporate patients’ electronic data and electronic prescriptions); nevertheless, more work has to be done, particularly in terms of the primary and secondary health sectors’ connectivity). Similar results were reported in countries with modern health information systems, such as the U.S. (88%) [33] and Spain (77%) [39], while lower results were reported in countries such as Yemen (52%) [40] and Poland (65%) [41], where primary care services are not supported by an information system.

The domains of “Work Pressure and Pace” (46%) and “Leadership Support” (62%) had the least favorable answers. This was mostly due to a longstanding problem in Greece’s healthcare system: a shortage of nurses. According to the WHO [42], in Greece, there are 3.6 nurses per 1000 population, compared to 9.1 nurses per 1000 in the OECD. Switzerland, Norway, and Denmark all have more than 16 nurses per 1000 residents, with Switzerland demonstrating the highest ratio with 17.4 nurses per 1000 population.

Studies in the USA highlight understaffing of physicians in primary care. A report by the Association of American Medical Colleges [43] estimated that by 2032, the USA would face a shortage of up to 55,200 primary care doctors, compared with about 480,000 primary care physicians in the USA in 2019. Also, people in the United States are perplexed by primary care specialists, skeptical of their attempts to provide quality healthcare, and unable to connect primary care to science or technology. They show that generalist specialties face a lack of respect in academic circles, administrative responsibilities, restrictive appointment schedules, and brief visits that satisfy neither the patient nor the physician [43]. Respectively in Greece, several barriers were identified in terms of waiting time for appointments, physicians’ access to patient medical history, delivery of preventive services, and patient involvement in decision-making [44].

This study was conducted in the 1st health region of Greece, Attica, which may be a limitation for results generalization. Another limitation of the study is that the initial questionnaire was conducted out of pandemic crisis, and the ideal situation would be to conduct the study in a period out of crisis; however, in our opinion it is worthwhile to investigate patient safety and quality issues from the staff’s perspective, even in the beginning of the pandemic period in Greece.

Second, the findings of this study reflect the perspectives of health professionals working in primary care settings, not administrative or technical employees. Finally, no attempt was made to compare the validity of the units’ evidence to other evaluation reports, such as interviews or record reviews.

Nevertheless, the results obtained in this research contribute to the dissemination of knowledge on the subject, as there is still little data in the literature. It is noteworthy that this study of psychometric validation is unprecedented in Greece, setting a starting point for future investigations that can be performed in other Greek regions.

## 5. Conclusions

The survey on patient safety Culture in primary care presented valid and reliable psychometric properties when applied for the first time in Greek PHU’s. Patient safety culture was positive in most tool domains.

Most highly rated dimensions included: teamwork, organizational learning, patient care tracking and follow-up, overall perception of patient safety and quality, and leadership support for patient safety. Slightly worse results referred to: staff training, communication about error, office processes, and standardization. The worst ratings from respondents referred to communication openness, leadership support, work pressure, and pace.

Our study was an important first step in examining perceptions of different professions. Hopefully, this leads to more attention and research in this area of healthcare. We believe it is necessary to conduct further research, desirably with mixed methods to further explore attitudes towards patient safety and identify specific needs for improvements.

The obtained results are fundamental for the tool application in studies that intend to assess patient safety culture in PHUs in different regions of the country.

## Figures and Tables

**Figure 1 medicines-08-00042-f001:**
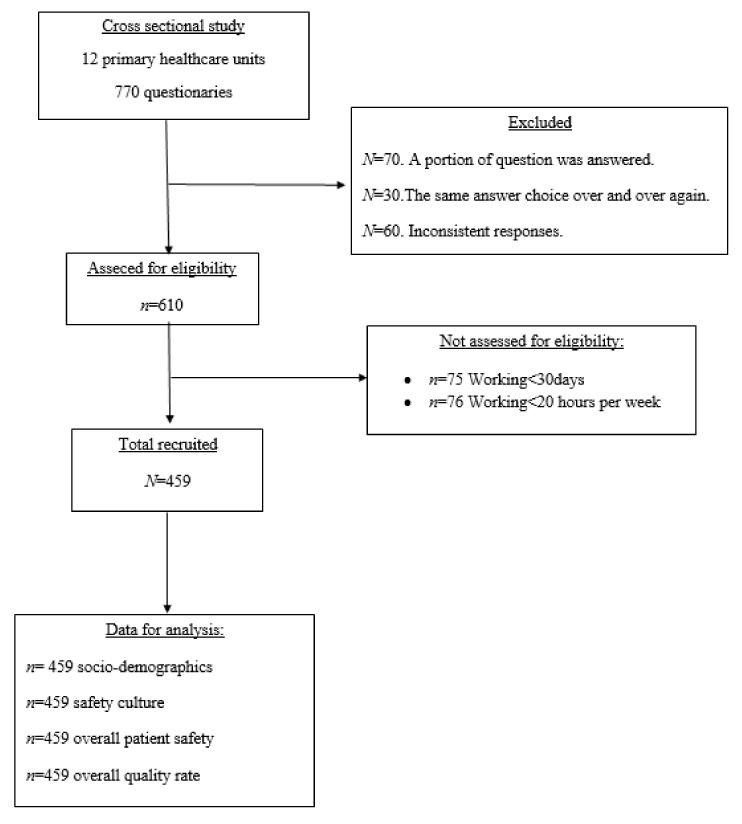
Flow chart of study sample.

**Table 1 medicines-08-00042-t001:** Definitions per safety culture dimension and the related items regarding MOSPSC * survey tool.

Patient Safety Culture Dimensions	Definition	Items
Communication about error	Staff are willing to report mistakes they observe and do not feel like their mistakes are held against them, and providers and staff talk openly about office problems and how to prevent errors from happening	4 (D7R, D8, D11,D12)
Communication Openness	Providers in the office are open to staff ideas about how to improve office processes, and staff are encouraged to express alternative viewpoints and do not find it difficult to voice disagreement	4 (D7R, D8, D11, D12)
Office Processes and Standardization	The office is organized, has an effective workflow, has standardized processes for completing tasks, and has good procedures for checking the accuracy of work performed	4 (C8R, C9, C12R, C15)
Organizational Learning	The office has a learning culture that facilitates making changes in office processes to improve the quality of patient care and evaluates changes for effectiveness.	3 (F1, F5, F7)
Overall Perceptions of Patient Safety and Quality	The quality of patient care is more important than getting more work done, office processes are good at preventing mistakes, and mistakes do not happen more than they should	4 (F2, F3R, F4R, F6R)
Owner/Managing Partner/Leadership Support for Patient Safety	Office leadership actively supports quality and patient safety, places a high priority on improving patient care processes, does not overlook mistakes, and makes decisions based on what is best for patients.	4 (E1R, E2R, E3, E4R)
Patient Care Tracking/Follow up	The office reminds patients about appointments, documents how well patients follow treatment plans, follows up with patients who need monitoring, and follows up when reports from an outside provider are not received.	4 (D3, D5, D6, D9)
Staff Training	The office provides staff with effective on-the-job training, trains staff on new processes, and does not assign staff tasks they have not been trained to perform.	3 (C4, C7, C10R)
Teamwork	The office has a culture of teamwork, mutual respect, and close working relationships among staff and providers	4 (C1, C2, C5, C13)
Work Pressure and Pace	There are enough staff and providers to handle the patient load, and the office work pace is not hectic	4 (C3R, C6R, C11, C14R)

* MOSPSC = Medical Office on Patient Safety Culture.

**Table 2 medicines-08-00042-t002:** Type of medical services provided at 22 primary healthcare units and demographic characteristics of staff completing the survey.

Services of PHU *		*n* (67)	(%)
	Medical	35	52
	Microbiology	22	33
	Radiology	10	15
Respondents		*N* (459)	(%)
Staff position	Nurses	190	41
	Physicians	95	21
	Midwifes	80	17
	Administration	55	12
	Other	39	9
Female		312	68
Age of respondents	<30 years old	50	11
	30-39 years old	152	33
	40-49 years old	124	27
	≥50 years old	116	25
	No answer for age	17	4
Primary work	Medical	316	69
	Radiology	45	10
	Microbiology laboratory	43	9
	Clerical staff	55	12
Length of time in PHU	1 to 5 years	115	25
	6 to 10 years	160	35
	More than 10	184	40
Working hours per week	25 to 32 h per week	72	16
	33 to 40 h per week	350	76
	≥41 h per week	37	8

* PHU = primary healthcare unit.

**Table 3 medicines-08-00042-t003:** Mean scores and standard deviation of the items. Factor loadings regarding patient safety culture.

Items	Mean	SD	Composite 1	Composite 2	Composite 3	Composite 4	Composite 5	Composite 6	Composite 7	Composite 8
D11	4.22	1.106	0.94							
C2	4.26	1.067	0.94							
D8	4.21	1.081	0.94							
C5	4.24	1.101	0.94							
D12	4.18	1.051	0.91							
F5	4.21	1.108	0.90							
C13	4.2	1.045	0.89							
F7	4.18	1.152	0.89							
D1	4.08	1.183	0.84							
D2	4.06	1.154	0.80							
F1	4.01	1.084	0.68							
D7R	3.83	1.168	0.66							
E3	4.07	1.252	0.57							
E2R	3.85	1.168		0.87						
D10R	3.59	1.345		0.86						
E4R	3.67	1.368		0.83						
D4R	3.8	1.305		0.77						
C12R	4.02	1.136			0.86					
C8R	4.2	1.063			0.84					
C15	3.83	1.268			0.77					
C9	3.81	1.071			0.77					
C3R	3.49	1.443				0.85				
E1R	3.38	1.432				0.84				
C14R	3.84	1.318				0.74				
F4R	3.93	0.899					0.95			
F3R	3.95	0.877					0.95			
F6R	3.94	1.152					0.65			
C7	3.83	1.046						0.88		
C4	3.75	0.97						0.88		
C10R	3.57	1.272						0.73		
D5	3.97	1.111							0.97	
D6	4.04	1.098							0.96	
C11	3.59	1.307								0.86
C6R	3.39	1.357								0.81

**Table 4 medicines-08-00042-t004:** Cronbach’s alpha and characteristics of the composites after factor analysis.

MOSPSC Factor Analysis			G-MOSPSC * Factor Analysis			
Composites	Items	Cronbach’s α American Data	Cronbach’s α Greek Data	Composites	Items	Cronbach’s α
1. Teamwork	4	0.83	0.82	1. Teamwork (C2, C5, C13) + 3. Organizational learning (F1, F5, F7) + 6. Owner/Managing Partner/Leadership Support for Patient Safety (E3) + 7. Communication About Error + 8. Communication Openness (D1, D2)	13	0.96
2. Patient Care Tracking/Follow up	4	0.78	0.74	2. Patient Care Tracking/Follow up (D5, D6)	2	0.964
3. Organizational learning	3	0.82	0.80	4. Overall Perceptions of Patient Safety and Quality (F4R, F3R, F6R)	3	0.834
4. Overall Perceptions of Patient Safety and Quality	4	0.74	0.70	5. Staff Training *	3	0.790
5. Staff Training	3	0.63	0.72	6. Owner/Managing Partner/Leadership Support for Patient (E2R, E4R) + 8. Communication Openness (D4R, D10R)	4	0.890
6. Owner/Managing Partner/Leadership Support for Patient Safety	4	0.76	0.71	6. Owner/Managing Partner/Leadership Support for Patient Safety(E1R) + 10. Work Pressure and Pace(C3R, C14R)	3	0.838
7. Communication About Error	4	0.80	0.72	9. Office Processes and Standardization *	4	0.839
8. Communication Openness	4	0.81	0.72	10. Work Pressure and Pace (C11, C6R)	2	0.692
9. Office Processes and Standardization	4	0.78	0.71			
10. Work Pressure and Pace	4	0,76	0.72			

* GMOSPSC = Greek version of Medical Office on Patient Safety Culture.

**Table 5 medicines-08-00042-t005:** Composite Measure Results. Average Percent Positive Response.

Patient Safety Culture Composite Measures	Average (%) Positive Response	
	AHRQ Database	This Study
Patient Care Tracking/Follow up	88	80
Teamwork	86	82
Organizational Learning	81	80
Overall Perceptions of Patient Safety and Quality	80	77
Staff Training	75	70
Communication About Error	74	70
Communication Openness	72	64
Office Processes and Standardization	70	67
Owner/Managing Partner/Leadership Support for Patient Safety	69	62
Work Pressure and Pace	49	46

**Table 6 medicines-08-00042-t006:** Overall rating on quality and safety in current study (*n* = 459) compared with AHRQ Database (*n* = 18,396).

		This Study (%)	AHRQ Database 2020
		Excellent/Very Good	Excellent/Very Good
Overall rating on quality issues	Patient centered	75	71
	Effective	75	71
	Timely	65	56
	Efficient	63	62
	Equitable	85	84
Overall rating on patient safety		70	68

## Data Availability

Not Applicable.

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
