# Peer review of "Psychometric Properties of the Greek Version of the Medical Office on Patient Safety Culture in Primary Care Settings"

_medicines, 2021, doi:10.3390/medicines8080042_

Round 1

Reviewer 1 Report

When reviewing scientific papers for publication, I usually start with a general overview in terms of a structure, abstract, literature review, methodology, findings of the research, discussion, conclusions, as well as limitations of the study.

In the assessment of the paper submitted for the review, I specifically focused on the discussed issues, applied research methods and the scope of analysis of research results, as well as substantive content of the article and its structure.

I do however have some remarks and feedback on the study.

1.Introduction can be more detailed, including the medical errors, the specific year and so on.

2.Supplementary conclusions, including study scope and limitations.

Author Response

Dear Editor,

Thank you again for giving us the opportunity to submit a revised draft of the manuscript with ID: medicines-1282074 in the Journal medicines. We appreciate the time and effort that you and the reviewers dedicated to providing feedback on our manuscript and are grateful for the insightful comments on and valuable improvements to our paper. We have tried to incorporate the suggestions made by the reviewers. Those changes are highlighted within the manuscript. Please see below, in blue, for a point-by-point response to the reviewers’ comments and concerns. Page numbers and lines refer to the revised manuscript.

Reviewer’s Comments to the Authors:

Reviewer #1

  1. Comment from reviewer #1

“Introduction can be more detailed, including the medical errors, the specific year and so on.”

Author response:

Thanks to the reviewer for the suggestion of optimizing the Introduction section with more evidence about medical malpractice in primary health care.

We believe we have addressed this issue by adding three paragraphs (lines 35-59) associated with 12 citations highlighting the significance of medical error in primary care.

  1. Comment from reviewer #1

“Supplementary conclusions, including study scope and limitations.”

Author response:

Thanks to the reviewer for giving us the opportunity to clarify this issue. The main aim of the study is the adaption and validation of MSOPSC tool in Greek language presenting the results of factor analysis. However, it is correct to observe that additional conclusion, scopes and limitations may arise.

We refer to them on lines:

  • 96-98 (Introduction section),
  • 323-326 (limitations) and
  • 335-341 (conclusions)

Reviewer 2 Report

Thank you for the opportunity to review this paper. This is an interesting and well written manuscript on matter of evaluation of primary care settings in terms of safety and quality, which is a growing concern worldwide. The topic of this manuscript is relevant and interesting, specially because it is placed in Greece, which had to make some reforms in primary care in the last decade. However, the manuscript needs some improvements.

Introduction - Introduction was mostly well written, covered the main background points and led up to the aim of the study. However, the most references are 10 or more years old. Therefore, I suggest authors to do new literature search because I believe that article would benefit of adding some newer data. Also, as all of these questionnaires were developed in the USA, it would be interesting to insert a few sentences about the differences in primary health care in the USA and Greece. 

Methods

Methods section is well described and it enables reproducibility. You have put exclusion criteria in the manuscript (Lines 120-126) and Fig.1 and it is easy to follow the text. However, it seem unclear what do you mean with  "contradictory answers", how did you conclude that someone belonged to that group, so I suggest you to put some explanation into the text because it is 160 questonnaries (160/770= 20.78%).

Disscusion

It is not usual to have tables in disscusion section. Therefore, I suggest authors to move the Table 6 in results section, as Table 5. also presents a similar comparison and it is already part of results.

The study has a huge limitation and it is already written as a limitation. Pandemic crisis was really not a good timing to conduct the study about quality in primary health care but you can't change that. But you should more clearly point it out what is actually so new and significant in the article that it should be interesting to the future readers? Therefore, I suggest authors to improve manuscript with adding some comparisons of the primary health care systems in different EU countries and to try explain differences in the results.

Author Response

Manuscript ID: healthcare-1279026

Response to Reviewers

Dear Editor,

Thank you again for giving us the opportunity to submit a revised draft of the manuscript with ID: medicines-1282074 in the Journal medicines. We appreciate the time and effort that you and the reviewers dedicated to providing feedback on our manuscript and are grateful for the insightful comments on and valuable improvements to our paper. We have tried to incorporate the suggestions made by the reviewers. Those changes are highlighted within the manuscript. Please see below, in blue, for a point-by-point response to the reviewers’ comments and concerns. Page numbers and lines refer to the revised manuscript.

Reviewer’s Comments to the Authors:

Reviewer #2

  1. Comment from Reviewer #2:

“Introduction was mostly well written, covered the main background points and led up to the aim of the study. However, the most references are 10 or more years old. Therefore, I suggest authors to do new literature search because I believe that article would benefit of adding some newer data. Also, as all of these questionnaires were developed in the USA, it would be interesting to insert a few sentences about the differences in primary health care in the USA and Greece.”

Author response:

Thanks to the reviewer for the suggestion of optimizing the Introduction section with more recent references. We believe we have addressed this issue by adding three paragraphs (lines 35-59) associated with 12 citations highlighting the significance of medical error in primary care.

In response to the reviewer's comment about differences in primary health care in the United States and Greece, we reorganized the Discussion section in order to focus on the differences and similarities of our study's findings with those of other similar studies looking at specific issues of health systems in different countries.

We believe that we have addressed this issue at lines 272-316  of the Discussion section.

  1. Comment from Reviewer #2:

“Methods section is well described and it enables reproducibility. You have put exclusion criteria in the manuscript (Lines 120-126) and Fig.1 and it is easy to follow the text. However, it seems unclear what do you mean with  "contradictory answers", how did you conclude that someone belonged to that group, so I suggest you to put some explanation into the text because it is 160 questonnaries (160/770= 20.78%)..”

Author response:

Thanks to the reviewer for his remark. Maybe the term “contradictory” is not the suitable one.

We replaced this term with the sentences at lines 153-157 in order to be in accordance with what is described in figure 1.

  1. Comment from Reviewer #2:

“It is not usual to have tables in disscusion section. Therefore, I suggest authors to move the Table 6 in results section, as Table 5. also presents a similar comparison and it is already part of results.

The study has a huge limitation and it is already written as a limitation. Pandemic crisis was really not a good timing to conduct the study about quality in primary health care but you can't change that. But you should more clearly point it out what is actually so new and significant in the article that it should be interesting to the future readers? Therefore, I suggest authors to improve manuscript with adding some comparisons of the primary health care systems in different EU countries and to try explain differences in the results.”

Author response:

 Thanks to the reviewer for his apposite remark. Table 6 removed to the Results section.

Concerning the limitations of the study, we truly believe that they are overcome for the following reasons:

  • Factor analysis revealed that the Greek version of the MSOPSC tool is reliable and validated, so the MSOPSC tool may be implemented in Greek primary care settings.
  • This study showed that primary care professionals in Greece are rather positive in their opinion of their patient safety culture
  • Our study is the first step in examining perceptions of different professions.
  • It’s the starting point for future researchers to conduct further research, desirably with mixed methods to further explore attitudes towards patient safety and identify specific needs for improvements.  

The relevant changes in the manuscript concerning these remarks have been made in the following lines: 337-345

Round 2

Reviewer 2 Report

The manuscript has been significantly improved by adding all those changes. I believe that paper is now much more interesting to the future readers and quality of presentation has been raised to a higer level. Therefore, I would like to congratulate the authors on excellent work.